# P-Cadherin Regulates Intestinal Epithelial Cell Migration and Mucosal Repair, but Is Dispensable for Colitis Associated Colon Cancer

**DOI:** 10.3390/cells11091467

**Published:** 2022-04-27

**Authors:** Nayden G. Naydenov, Susana Lechuga, Ajay Zalavadia, Pranab K. Mukherjee, Ilyssa O. Gordon, David Skvasik, Petra Vidovic, Emina Huang, Florian Rieder, Andrei I. Ivanov

**Affiliations:** 1Department of Inflammation and Immunity, Lerner Research Institute, Cleveland Clinic Foundation, Cleveland, OH 44195, USA; naydenn@ccf.org (N.G.N.); lechugs@ccf.org (S.L.); mukherp2@ccf.org (P.K.M.); skvasid@ccf.org (D.S.); pvidovic22@jcu.edu (P.V.); riederf@ccf.org (F.R.); 2Imaging Core, Lerner Research Institute, Cleveland Clinic Foundation, Cleveland, OH 44195, USA; zalavaa@ccf.org; 3Department of Pathology, Robert J. Tomsich Pathology and Laboratory Medicine Institute, Cleveland Clinic Foundation, Cleveland, OH 44195, USA; gordoni@ccf.org; 4Department of Surgery, UT Southwestern Medical Center, Dallas, TX 75390, USA; emina.huang@utsouthwestern.edu; 5Department of Gastroenterology, Hepatology and Nutrition, Digestive Diseases and Surgery Institute, Cleveland Clinic Foundation, Cleveland, OH 44195, USA

**Keywords:** epithelial cells, cell migration, mucosal restitution, colon cancer, cell-cell adhesions, cadherins

## Abstract

Recurrent chronic mucosal inflammation, a characteristic of inflammatory bowel diseases (IBD), perturbs the intestinal epithelial homeostasis resulting in formation of mucosal wounds and, in most severe cases, leads to colitis-associated colon cancer (CAC). The altered structure of epithelial cell-cell adhesions is a hallmark of intestinal inflammation contributing to epithelial injury, repair, and tumorigenesis. P-cadherin is an important adhesion protein, poorly expressed in normal intestinal epithelial cells (IEC) but upregulated in inflamed and injured mucosa. The goal of this study was to investigate the roles of P-cadherin in regulating intestinal inflammation and CAC. P-cadherin expression was markedly induced in the colonic epithelium of human IBD patients and CAC tissues. The roles of P-cadherin were investigated in P-cadherin null mice using dextran sulfate sodium (DSS)-induced colitis and an azoxymethane (AOM)/DSS induced CAC. Although P-cadherin knockout did not affect the severity of acute DSS colitis, P-cadherin null mice exhibited faster recovery after colitis. No significant differences in the number of colonic tumors were observed in P-cadherin null and control mice. Consistently, the CRISPR/Cas9-mediated knockout of P-cadherin in human IEC accelerated epithelial wound healing without affecting cell proliferation. The accelerated migration of P-cadherin depleted IEC was driven by activation of Src kinases, Rac1 GTPase and myosin II motors and was accompanied by transcriptional reprogramming of the cells. Our findings highlight P-cadherin as a negative regulator of IEC motility in vitro and mucosal repair in vivo. In contrast, this protein is dispensable for IEC proliferation and CAC development.

## 1. Introduction

Intestinal epithelium is a constantly rejuvenating tissue with the colonic mucosal lining being shed and regenerated every 3–5 days. The intestinal rejuvenation involves proliferation of stem cells residing at the base of the crypt with subsequent cell movement to the luminal surface, lined by well-polarized intestinal epithelial cells (IEC) [1,2]. Such epithelial renewal becomes markedly abnormal in the chronically inflamed intestinal mucosa of patients with inflammatory bowel diseases (IBD) and particularly in ulcerative colitis (UC) [1,3,4]. Multiple factors contribute to abnormal epithelial renewal including excessive IEC death, inappropriate immune cell infiltration, secretion of various pro-inflammatory mediators, and altered microbiota composition [3,5,6,7]. The deranged regeneration of the injured colonic mucosa involves a partial de-differentiation of IEC and their return to an oncofetal state [8]. This phenotypic plasticity is likely to represent an adaptive response aimed at generating highly proliferative and motile cells to accelerate mucosal wound repair. However, extensive and deregulated epithelial plasticity could become poorly reversible leading to the impaired restoration of the intestinal epithelial barrier, or in most severe cases to the oncogenic transformation and development of colitis associated colon cancer (CAC).

A crucial consequence of the increased phenotypic plasticity of the inflamed intestinal epithelium is profound alterations in intercellular adhesions. The IEC create a tight epithelial barrier by assembling the apical junctional complex, comprising adherens junctions (AJs) and tight junctions (TJs) [9,10,11,12]. The AJs are crucial drivers of epithelial adhesions responsible for initiation of cell-cell contacts and assembly of other junctional complexes [10,11]. Furthermore, AJs are known coordinators of epithelial cell responses during collective cell migration and wound healing [13,14]. E-cadherin is a principal adhesive component of AJs that plays several important roles, including regulation of the epithelial barrier, promoting cell differentiation and limiting proliferative responses via contact-dependent inhibition of cell division [10,11,15]. A characteristic feature of the epithelial plasticity in different diseases is a so called ‘cadherin switch’ manifested by the replacement of E-cadherin with other members of the cadherin superfamily [16,17,18]. The most notable example is downregulation of E-cadherin and upregulation of N-cadherin levels during the epithelial to mesenchymal transition at late stages of tumorigenesis [19,20].

A non-canonical cadherin switch has been described in the colonic mucosa of IBD patients. It involves downregulation of E-cadherin level accompanied by a marked expressional upregulation of a homologous adhesion protein, P-cadherin [18,21,22]. Upregulation of P-cadherin expression appears to be a prominent feature of both UC and CD mucosa, especially in dysplastic ulcerated tissues [23,24,25,26]. Furthermore, increased P-cadherin levels were observed in colorectal cancer [27,28,29,30] and highlighted as an early event in hyperplastic and dysplastic transformation in the colon [31]. The functional consequence of the E-cadherin to P-cadherin switching in inflamed and dysplastic colonic mucosa remain poorly understood. Despite high structural similarity, E-cadherin and P-cadherin display different adhesion properties [32] and regulate different epithelial signaling events [16,18,33]. In contrast to E-cadherin, P-cadherin was shown to have tumor-promoting activities in different types of cancer that involve stimulation of tumor growth and metastasis [34,35,36,37,38]. This adhesion protein has been most extensively studied in mammary epithelium and breast cancer where P-cadherin was implicated in various aspects of tumor development including regulation of breast cancer stem cells [39,40,41]. Previous studies addressing the involvement of P-cadherin in colon cancer cell adhesion and migration reported conflicting cell-specific phenotypes and did not provide mechanistic insights into P-cadherin modulation of IEC motility [42,43,44]. Furthermore, functional roles of P-cadherin in the regulation of intestinal mucosal restitution and colonic tumorigenesis in vivo remain poorly understood.

In this study, we investigated the roles of P-cadherin in IEC motility, mucosal repair, and colonic tumorigenesis. We found that P-cadherin suppresses collective epithelial cell migration in vitro and mucosal restitution in vivo without having significant effects on IEC proliferation and tumor growth. P-cadherin appears to control multiple molecular pathways responsible for cell migration in IEC monolayers. Among these pathways, activity of the Src family kinases, Rac1 small GTPase and non-muscle myosin II (NM-II) motors are critical for P-cadherin dependent regulation of IEC motility.

## 2. Materials and Methods

### 2.1. Antibodies and Other Reagents

The following primary polyclonal antibodies (pAb) and monoclonal antibodies (mAb) were used to detect adhesion, cytoskeletal and signaling proteins: anti P-cadherin mAb (Cat# MAB 861; used for immunoblotting of human IEC), and P-cadherin pAb (Cat# AF 761; used for immunoblotting of mouse tissues) from R & D Systems (Minneapolis, MN, USA); anti P-cadherin mAb (Cat# 610227, used for human tissue staining), paxillin (Cat# 610568), β1 integrin (Cat# 610467), E-cadherin (Cat# 610181), p120 catenin (Cat# 610133), and β-catenin (Cat# 610153) mAbs were from BD Biosciences (San Jose, CA, USA). Anti-E-cadherin (Cat# ab40772) and anti-total actin (Cat# ab1801) rabbit pAbs were from Abcam (Cambridge, UK); anti-Src (Cat# 2123) and anti-phospho-cofilin S3 (Cat# 3313) rabbit mAbs, as well as anti-phospho-Src (Cat# 2101), total FAK (Cat# 3285), phospho-FAK Y397 (Cat# 8556), phospho-paxillin (Y118) (Cat# 6936), total MLC (Cat# 3672), β4 integrin (Cat# 4707), phospho-MLC S19 (Cat# 3671), di-phospho-MLC T18/S19 (Cat# 3674), and GAPDH (Cat# 2818) pAbs were from Cell Signaling Technology (Beverly, MA, USA); anti-cofilin (Cat# SAB2702206); anti-cortactin (Cat# 05-180) mouse mAbs and anti-phospho-cortactin Tyr421 (Cat# AB3852) pAb were from Millipore-Sigma (Burlington, MA, USA); anti-α2-integrin (Cat# A7629), α3-integrin (Cat# A17502) and αV-integrin (Cat# A19071) pAbs were from ABclonal Technology (Woburn, MA, USA); anti-NM-IIA (Cat# 909801) and NM-IIB (Cat# 909901) pAbs were from Biolegend (San Diego, CA, USA); anti-occludin (Cat# 13409) and α-catenin (Cat# 12831) pAbs were from ProteinTech (Rosemont, IL, USA); anti-claudin-7 pAb (Cat# 349100) was from Thermo-Fisher (Waltham, MA, USA). Horseradish peroxidase (HRP)-conjugated rabbit anti-goat, goat-anti-rabbit and goat anti-mouse secondary antibodies were acquired from Bio-Rad Laboratories (Hercules, CA, USA). OmniMap anti-Rabbit HRP (Cat# 05269679001) and OmniMap anti-Mouse HRP (Cat# 05269652001) were from Roche (Basel, Switzerland). Opal 620 (Cat# FP1495001) and Opal 690 (Cat# FP1497001) fluorophores were from Akoya Biosciences (Marlborough, MA, USA). Both PP2 and EHT 1864 were purchased from Tocris Bioscience (Bristol, UK). Azoxymethane (AOM) and blebbistatin were obtained from Millipore-Sigma. All other reagents of the highest grade were obtained from either Thermo-Fisher, or Millipore-Sigma. 

### 2.2. Animals

P-cadherin knockout (KO) mice on C57Bl/6J background (JAX stock #003180) [45] were obtained from Jackson Laboratory (Bar Harbor, ME, USA). The animals were bred as heterozygotes and P-cadherin null and wild type littermates were used in the experiments whenever possible. The animal colony was maintained under pathogen-free conditions in the vivarium of the Lerner Research Institute of Cleveland Clinic. The mouse room was on a 12 h light/dark cycle and standard mouse chow and tap water were available, ad libitum. All procedures were conducted according to animal research protocols approved by the Lerner Research Institute Animal Care and Use Committee in compliance with the National Institutes of Health Animal Care and Use Guidelines.

### 2.3. Induction and Characterization of Dextran Sulfate-Induced Colitis and Post-Colitis Recovery

Experimental colitis and post-colitis recovery was induced in 8–10-week-old P-cadherin null and control mice by administering a 3% (*w*/*v*) solution of dextran sulfate, sodium salt (DSS, Thermo-Fisher Scientific), in drinking water, ad libitum. Vehicle-treated animals received tap water. Roughly equal numbers of male and female mice were used in each experimental group. At the beginning of colitis experiments, mice weighed 18–25 g, with no significant difference between the body masses of mice of different genotypes. Animals were weighed and monitored for symptoms of gastrointestinal disorder daily. The disease activity index was calculated by averaging numerical scores of body weight loss, stool consistency, and intestinal bleeding, as previously described [46,47]. With regards to body weight, no weight loss was scored as 0, loss of 1–5% of body weight from the beginning of the experiment was scored as 1, whereas 5–10% and 10–15% body weight loss was scored as 2 and 3, respectively. Body weight loss more than 15% was scored as 4. For stool consistency, a well-formed pellet was scored as 0, a soft and semi-formed stool was scored as 2, and a liquid stool or diarrhea was scored as 4. For intestinal bleeding, no blood was scored as 0, a hemoccult-positive stool was scored as 2, and gross rectal bleeding was scored as 4. During acute DSS colitis experiments, the animals were euthanized on day 7 of DSS administration. For the post-DSS recovery study, DSS was replaced with tap water on day 7 and animals were allowed to recover for an additional eight days. Harvested distal colonic segments were fixed in 10% formaldehyde solution, paraffin embedded, sectioned, and stained with hematoxylin and eosin (H and E). The H and E stained sections of distal colon were examined, “blind”, by a gastrointestinal pathologist, and the tissue injury index was calculated as previously described [48]. The index represents the sum of individual scores reflecting the degree and extent of mucosal inflammation, leukocyte infiltration, and epithelial injury. 

### 2.4. AOM/DSS Induced CAC

The AOM/DSS model of colitis was carried out as described previously [49,50]. Animals were 8–10 weeks old at the beginning of the study and roughly equal numbers of male and female mice were included in each experimental group. P-cadherin KO mice and their wild type controls were given a single intraperitoneal injection of AOM (10 mg/kg body weight). Three days after the AOM injection, the mice were treated with 2.5% DSS in their drinking water for five days, followed by two weeks of recovery (tap water only); this treatment was repeated for two additional 19 day cycles. At the end of the third cycle, animals were anesthetized, subjected to endoscopic examination, euthanized and their colons were dissected and examined for tumor formation. The H and E stained Swiss-rolled colon sections were scored by an IBD pathologist for adenocarcinoma in a manner blinded to the experimental groups according to literature [51]. In brief, the criteria were listed as score 0, normal; 1, low-grade dysplasia; 2–3, high-grade dysplasia; 4–5, intramucosal adenocarcinoma and invasive adenocarcinoma.

### 2.5. Mouse Colonoscopy

Mouse colonoscopy was performed using a small animal endoscopy system (Karl Storz Endoscopy America, El Segundo, CA, USA) equipped with TH102 digital camera and medical video recorder. Mice were anesthetized with 2% isoflurane and oxygen (1 L/min). The colon was washed with phosphate buffered saline to cleanse the bowel. Air was carefully insufflated into the colon using a hand insufflation pump to allow full visualization of the colonic surface. Endoscopic images and movies were viewed using a standard endoscopy monitor (Karl Storz) and saved for later analysis.

### 2.6. Cell Culture

The HCA-7 Colony 29 (referred hereafter as HCA-7) human colonic epithelial cells were obtained from the European Collection of Authenticated Cell Cultures (Cat# ECACC 02091238, Porton Down, UK). The SK-CO15 human colonic epithelial cells [52] were a gift from Dr. Enrique Rodriguez-Boulan (Weill Medical College of Cornell University, New York, NY, USA). The SW620 human colon cancer cells (Cat# CCL-227) were obtained from the American Type Culture Collection (Manassas, VA, USA). The HCA-7 and SK-CO15 cells were cultured in DMEM medium supplemented with 10% fetal bovine serum (FBS), HEPES, non-essential amino acids, and penicillin-streptomycin antibiotic. SW620 cells were cultured in Leibovitz’s L-15 Medium. All cultured cells were mycoplasma free according to the PCR-based mycoplasma detection assay (PromoCell, Heidelberg, Germany). The IEC were plated on either 6-well plates, or collagen-coated coverslips for biochemical and functional/imaging studies, respectively.

### 2.7. Human Intestinal Tissue Samples and Isolation of Primary Colonic Epithelial Cells

Deidentified sections of CAC were obtained from surgical resections performed at the Cleveland Clinic Digestive Diseases and Surgery Institute. Sections of non-affected colons from patients with sporadic or metastatic colorectal cancer, taken at least 10 cm away from the cancer site, served as normal controls. Deidentified colonic surgical resection specimens obtained from IBD patients with active Crohn’s Disease (CD), inflamed ulcerative colitis (UC) and non-IBD controls (diverticulosis, chronic constipation, or non-affected margins of resected colorectal cancer) were provided by the Biorepository Core of the Cleveland Digestive Diseases Research Core Center. Colonic epithelial cells were isolated from the tissue segments using a previously described protocol [53]. Briefly, resected samples were cut into approximately 1.5 cm long strips and washed with HEPES-buffered Hanks’ balanced salt solution without calcium and magnesium (HBSS-). Colonic epithelial cells were isolated by sequential incubation of the tissue strips in HBSS-containing 10 mM dithiothreitol for 30 min followed by incubation in HBSS-containing 1 mM EDTA (pH 7.4) under constant agitation for 3 h at room temperature. Detached epithelial cells were collected by centrifugation for 10 min at 3000 rpm. Cells were washed once in ice-cold HBSS- and lysed with RIPA buffer (20 mM Tris, 150 mM NaCl, 2 mM EDTA, 2 mM EGTA, 1% sodium deoxycholate, 0.1% SDS, 1% Triton-X100, pH 7.4) supplemented with phosphatase inhibitor cocktails 2 and 3 (1:200), protease inhibitor cocktail (1:100) and 0.5 mg/mL Pefablock (all from Millipore-Sigma). Colonic epithelial cell lysates were subjected to immunoblotting analysis as described below. The study conforms to the US Federal policy for protection of human subjects and was approved by the Institutional Review Board of Cleveland Clinic Foundation as a minimal risk investigation. 

### 2.8. Generation of P-Cadherin Knockout and Overexpressing Cell Lines

P-cadherin-knockout HCA-7 and SK-CO15 cell lines were constructed using the CRISPR/Cas9 V2 system. Single guide RNAs (sgRNA) were designed with a CRISPR design web site (http://crispr.mit.edu/ (accessed on 1 February 2020)), provided by the Feng Zhang Lab (McGovern Institute, Massachusetts Institute of Technology, Boston, MA, USA). Their sequences are as follows: sg1-TCTACAGCGAAGACACCCTC, sg3 GTTGTTGTTGAATAAGCCAC and control sg-CACCGGACCGGAACGATCTCGCGTA. P-cadherin and control sgRNAs were cloned into a BbsI restriction site of the lentiCRISPR v2 vector (Cat# 52961, Addgene Watertown, MA, USA) and confirmed by sequencing. Lentiviruses were produced in HEK293T cells transfected with packaging plasmids pCD/NL-BH*DDD (Cat# 17531, Addgene) and pLTR-G (Cat# 17532, Addgene) using a TransIT-293 transfection reagent (Mirus Bio, Madison, WI, USA). Stable P-cadherin-depleted IEC cells were generated by transducing HCA-7 and SK-CO15 cells with lentiviruses containing P-cadherin sgRNAs and subsequent puromycin selection (5 μg/mL) for 7 days. Control IEC were generated via transduction with the control sgRNA containing lentivirus and puromycin selection.

Stable SW620 cell lines were engineered to express either full-length P-cadherin, or control pcDNA. P-cadherin plasmid (Cat# 47502) and control pcDNA plasmid (Cat# 128034) were obtained from Addgene. Cells were transfected using Lipofectamine 2000 (Thermo-Fisher Scientific). Forty-eight hours (48 h) after transfection, fresh medium containing G418 antibiotic (0.5 mg/mL) was added. After 7 days of G418 selection, cells were cultured in Leibovitz’s L-15 medium with 10% FBS and P-cadherin expression was analyzed by immunoblotting.

### 2.9. Immunoblotting Analysis

Confluent IEC monolayers were subjected to multiple wounding using a Cell Comb™ Scratch kit (Millipore-Sigma). Six hours post wounding cell were homogenized cells with RIPA cell lysis buffer supplemented with phosphatase inhibitor cocktails 2 and 3 (1:200), protease inhibitor cocktail (1:100) and 0.5 mg/mL Pefablock. To examine protein phosphorylation, wounded cell monolayers were homogenized using the PhosphoSafe Extraction Reagent (Millipore-Sigma) supplemented with protease inhibitor cocktail (1:100) and 0.5 mg/mL Pefablock. The homogenized samples were centrifuged, the resulting supernatant mixed with an equal volume of 2X SDS sample buffer (Bio-Rad Laboratories), then boiled. Total cell lysates were separated using SDS-polyacrylamide gel electrophoresis with 10–20 µg of total protein loaded into each well. The separated proteins were transferred onto nitrocellulose membranes by electroblotting. After transfer, membranes were sequentially incubated with primary and HRP-conjugated secondary antibodies, and the labeled proteins were visualized using standard enhanced chemiluminescence reagents (Millipore-Sigma) and X-ray films. Protein expression was quantified by densitometry (Epson Perfection V500 photo scanner) using ImageJ 1.52p software (National Institute of Health, Bethesda, MD, USA). Densitometric analysis was performed on three different immunoblots, each representing an independent experiment. Data are presented as normalized values assuming the expression level in control sgRNA-treated groups as 1.

### 2.10. Immunohistochemistry

Tissue samples of dissected colitis associated colon cancer and normal intestinal mucosa (diverticulosis and healthy margins of dissected tumors) were fixed in formalin and paraffin embedded. Immunohistochemistry staining of the tissue sections was performed using the Discovery ULTRA automated stainer from Roche Diagnostics. The antigen retrieval was performed using a Discovery CC1 Tris/borate/EDTA buffer (Cat# 06414575001, Roche). Antigen denaturing was performed using a Discovery CC2 citrate buffer (Cat# 05424542001, Roche). Tissue sections were incubated with primary antibodies (P-cadherin mAb, 1:100 and E-cadherin pAb 1:1500) for 40 min at 37 °C. The antibodies were visualized using the OmniMap anti-Mouse HRP and OmniMap anti-Rabbit HRP in conjunction with Opal 620 and Opal 690 fluorophores, respectively. The slides were counterstained with Spectral DAPI. Labeled slides were imaged on a Vectra Polaris multispectral slide and tissue imager (Akoya Biosciences). Multispectral slide scans were processed using inForm software (v2.5, Akoya Bioscience) to separate contributions from tissue autofluorescence and signal overlap. The separated image files resulting from the inForm unmixing workflow were stitched back together using the QuPath software [54] and saved as pyramidal OME-TIFF files for image analysis. A QuPath script along with a pixel classifier with appropriate threshold was used to identify the stained areas. The P-cadherin positive area was measured for the E-cadherin stained area to calculate the percentage of positive area for each sample.

### 2.11. Cell Migration Assays

Cells were plated on “Culture insert 3 well μ-Dish” (Cat# 80366, Ibidi USA, Fitchburg, WI, USA) and grown for 3 days until confluency. To initiate cell migration, the 3-well silicone gasket was removed, and the monolayers were supplied with fresh cell culture medium. The images of a cell-free area were acquired at the beginning and at the indicated times after removal of the internal insert, using a Keyence BZ-X710 microscope (Keyence, Osaka, Japan). The area of three different wounds was measured and the percentage of wound closure was calculated using Image J.

Time lapse imaging of wound healing in HCA-7 cell monolayers was performed using a Leica DMI6000 inverted microscope (Leica Microsystems, Wetzlar, Germany) equipped with adaptive focus, automated stage, environmental chamber, and ORCA Flash 4 camera using the LAS-X acquisition software (v3.6, Leica Microsystems). Phase contrast images at 10× were acquired at multiple places along the wound edges at 5 min intervals for 24 h. A random trees pixel classifier in QuPath was trained to identify the wound area not covered by cells. Detection results from the pixel classifier were used to calculate the wound area for every time point in the timelapse data for eight positions per sample. These area measurements were exported for calculation of the velocity of wound healing [55].

For individual cell migration assays, control and P-cadherin-knockout HCA-7 cells were plated on collagen-coated 35 mm coverglass bottom dishes (Cat# P35G-1.5-14-C, MatTek Corporation, Ashland, MA, USA) and allowed to attach for 12 h. Cells were imaged on the Leica DMI6000 inverted microscope (Leica Microsystems) equipped with automated stage, adaptive focus, environmental chamber and ORCA Flash 4 camera using the LAS-X acquisition software (v3.6, Leica Microsystems). Phase contrast images at 10× were acquired for multiple positions per dish at 5 min intervals for 24 h. An Image-Pro 10 software (Media Cybernetics, Rockville, MD, USA) was used to manually track 50 cells per sample. For visualization of the tracks, Rose plots for each sample were generated using a script in MatLab. The cell tracking data was analyzed to calculate speed, accumulated distance and directional ratio [56].

### 2.12. Extracellular Matrix Adhesion Assay

Cells were collected from plates using TrypLE^TM^ Express reagent (Thermo-Fisher Scientific), counted with a T20 automated cell counter (Bio-Rad laboratories), and resuspended in the appropriate cell culture medium. Ten thousand (10,000) HCA-7, SK-CO15 or SW620 cells were seeded per well on a 24-well plate coated with either rat collagen I or human fibronectin and allowed to adhere for 30 min at 37 °C. Unattached cells were aspirated and the wells were gently washed with HBSS buffer. The attached cells were fixed with methanol and stained using a DIFF stain kit (Electron Microscopy Sciences, Hatfield, PA, USA). Adherent cells were imaged using the Keyence BZ-X710 microscope. Three 20× field images were captured per each well. The number of adhered cells in each image was counted using ImageJ and averaged to yield a single data point.

### 2.13. Cell Spreading Assay

Cells were collected from plates using TrypLE^TM^ Express reagent and counted. Next, 3000 cells were seeded on rat collagen I-coated 24-well plates. Cells were allowed to attach and spread over the coated surface for 1 h and were then imaged using the Keyence BZ-X710 microscope. The surface area of spread cells was measured using the ImageJ program. In each experiment, 50 randomly selected cells were measured for every experimental group. 

### 2.14. Cell Proliferation Assay

For cell counting, daily samples were taken for a period of four consecutive days. Cells were trypsinized and aggregated cells were disrupted by gently pipetting. Cells were counted using automated cell counter as described above. 

### 2.15. Rac1 GTPase Activation Assay

Rac1 activity in control and P-cadherin knockout HCA-7 cells was determined by the enzyme-linked immunosorbent assay using a G-LISA activation assay kit (Cytoskeleton, Denver, CO, USA). The assay was carried out according to the manufacturer’s protocol, with an internal active Rac1 standard used as a positive control. Absorbance of the processed samples at 490 nm was measured using a microplate reader.

### 2.16. RNA Sequencing Analysis

Total RNA was extracted from either control, or two P-cadherin knockout HCA-7 cell lines, using the RNeasy Mini Kit (Qiagen, Hilden, Germany). The RNA quality was assessed using the Agilent bioanalyzer and samples with an RNA quality score of >9.0 were used in RNA-sequencing. The RNA concentrations were determined using Qubit^®^ 3.0 Fluorometer (Thermo-Fisher Scientific). The RNA-sequencing libraries were generated using Illumina TruSEQ kits following the manufacturer’s protocol and libraries were sequenced using an Illumina NovaSeq 6000 following Illumina reagents and protocols. Paired-ended 100 base pair reads were checked for quality with FastQC (v0.11.7) (http://www.bioinformatics.babraham.ac.uk/projects (accessed on 24 March 2022)) before alignment to the human genome (GRCh38.p13). Reads were aligned using an Rsubread package [57]. Overall, an average of 98.0% of reads aligned uniquely. Gene counts were determined by the number of uniquely aligned, unambiguous reads (Subread: featureCounts, v1.5.2) and annotated (GRCH38, Ensemble 99 release). On average, 70% of reads in each sample were successfully assigned. Raw counts were loaded into *R* (v3.6.2 and 4.1.2, R Core Team 2021 https://www.R-project.org (accessed on 24 March 2022)), and subsequent analyses were performed using standard packages. Counts were filtered to exclude transcripts that were expressed at low levels (counts per million reads mapped [CPM]  <  1) before performing differential expression (DE) analysis using R packages edgeR [58] and Limma [59] software. Differentially expressed genes (DEGs) with Benjamini–Hochberg adjusted *p*-value < 0.05 were considered statistically significant. Volcano plots were created using R package software EnhancedVolcano (https://github.com/kevinblighe/EnhancedVolcano (accessed on 24 March 2022)). Functional analysis of significant DEGs were performed using R package clusterProfiler (v3.14.3) software [60] to identify and plot enriched Gene Ontology terms and KEGG pathways. Enrichment plots were created using an EnrichmentMap application in Cytoscape software (v3.8.2, https://cytoscape.org/ (accessed on 24 March 2022)). The RNA sequencing data have been deposited in Gene Expression Omnibus. GEO accession number GSE199859. 

### 2.17. Statistics

All data is expressed as means ± standard error (SE) from three biological replicates. Statistical analysis was performed using a two-tailed unpaired Student *t*-test to compare results obtained with two experimental groups (i.e., P-cadherin immunolabeling in human CAC samples, mouse colitis and CAC studies and functional analysis of SK-CO15 and SW620 cells). When data from three experimental groups were compared (i.e., in control and two different P-cadherin knockout HCA-7 cell lines) a one-way ANOVA was used. If the ANOVA test showed significant differences, a Dunnett post-hoc test was used to compare the difference between the control and each P-cadherin-depleted group. *p* values < 0.05 were considered statistically significant. The statistical analysis was performed using Prism 9.10 software (GraphPad, La Jolla, CA, USA).

## 3. Results

### 3.1. P-Cadherin Expression Is Upregulated in the Colonic Mucosa of UC and CAC Patients

As previous studies have documented increased P-cadherin mRNA expression in the intestinal mucosa of IBD patients and in colon cancer tissues [24,25,26,27,28], we sought to determine if the protein level of this cadherin is upregulated in the colonic epithelium during IBD and CAC. Primary epithelial cells were isolated from resected colonic sections of patients with CD, UC and non-IBD controls. Immunoblotting analysis of total epithelial cell lysates revealed a significant upregulation of P-cadherin protein expression in the isolated colonic epithelium of UC patients (Figure 1A,B). Such P-cadherin induction was accompanied by decreased E-cadherin protein level, thereby revealing the cadherin switch in the UC colonic mucosa (Figure 1A,B). The expression of P-cadherin and E-cadherin proteins was also compared in tissue sections of CAC and non-involved normal colonic mucosa by immunofluorescence labeling. As we observed a patchy pattern of P-cadherin labeling throughout full thickness CAC sections and wanted to focus on epithelial cells, we measured the percentage of the E-cadherin positive area of the images containing the P-cadherin signal. P-cadherin labeling was barely detectable in the E-cadherin-positive normal colonic epithelium, but was markedly induced in the epithelial compartment of CAC samples (Figure 1C,D). Together, our data strongly suggests that P-cadherin protein is induced in the colonic epithelium of UC and CAC patients.

### 3.2. P-Cadherin Knockout in Mice Promotes Mucosal Restitution, but Does Not Affect Acute Colitis and CAC Growth

Given the observed upregulation of P-cadherin expression during intestinal inflammation and tumorigenesis, we sought to investigate whether this adhesion protein is involved in the development of experimental colitis and CAC in vivo using P-cadherin knockout mice. Mice with total knockout of P-cadherin are healthy and do not display major physiological abnormalities, except for precocious mammary gland development [45]. We confirmed the loss of P-cadherin protein in knockout animals by performing immunoblotting analysis of kidney and spleen tissue lysates expressing high levels of this cadherin [29] (Appendix A). Acute colonic inflammation was induced in P-cadherin KO mice and their wild-type littermates by exposing the animals to 3% DSS solution in drinking water for 7 days. No differences in acute colitis development were observed between P-cadherin KO and control animals based on similar dynamics of their body weight loss and the calculated diseases activity index (Figure 2A,B). Next, we investigated the effects of P-cadherin knockout on the post-colitis mucosal restitution. Mice were initially treated with 3% DSS for 7 days and were allowed to recover by replacing the DSS solution with tap water for additional 8 days. P-cadherin null mice demonstrated accelerated recovery after DSS-induced colitis as compared to the wild type controls. This was manifested by significantly faster restoration of animal body weight and more pronounced decline in the disease activity index during the recovery period (Figure 2C,D). The H and E staining of distal colonic sections revealed substantial epithelial regeneration 8 days after DSS withdrawal with residual areas of submucosal inflammation and immune cell infiltration (Appendix A). The extent of mucosal recovery was variable, with some animals completely regenerating colonic epithelium after DSS treatment. Due to this variability, the tissue inflammation and injury index did not show statistically significant differences between P-cadherin KO and control mice (Appendix A).

A classical AOM/DSS model was carried out to investigate the roles of P-cadherin in CAC development [49,50]. After AOM injection and three cycles of DSS administration, both P-cadherin null and controls mice developed colonic polyps/tumors observed by endoscopic examination on the distal colon (Figure 3A, arrows). However, the number of developed colonic tumors did not differ between P-cadherin-null mice and wild type controls (Figure 3B,C). Histological examination of dissected tumors revealed a predominantly multifocal dysplasia, including high grade dysplasia with only two of twenty-seven mice developing invasive adenocarcinoma. Figure 3D shows examples of high grade and low grade dysplasia in control and P-cadherin KO mice, respectively. Interestingly, the histopathology score of developed colonic tumors was significantly higher in P-cadherin null mice as compared to wild type littermates (Figure 3E). Together, the described animal experiments suggest that P-cadherin plays roles in regulating post-colitis restitution but has no effects on acute mucosal inflammation and inflammation-induced colonic tumor growth in vivo. 

### 3.3. P-Cadherin Knockout Accelerates IEC Wound Healing by Modulating Cell-Matrix Adhesion and Cell Spreading

The observed acceleration of post-colitis recovery of P-cadherin-null mice prompted us to investigate the roles of P-cadherin in modulating IEC migration. We used HCA-7 and SK-CO15 cells, which are well-differentiated human colonic carcinoma cell lines, known to be good models to study intestinal barrier integrity and repair [61,62,63,64]. Furthermore, we examined P-cadherin expression in a panel of different human colonic epithelial cell lines and found that HCA-7 and SK-CO15 cells have high levels of this protein (data not shown). CRISPR-Cas9-mediated gene editing was used to knockout P-cadherin in the selected IEC lines. Two HCA-7 stable cell lines generated using different P-cadherin sgRNAs and one SK-CO15 cell line with the most efficient P-cadherin knockout were used in the subsequent experiments (Figure 4A,B).

As healing of intestinal mucosal wounds is known to be driven by two critical mechanisms, epithelial cell migration and proliferation [65], we sought to investigate the roles of P-cadherin in regulating these cellular processes. Loss of P-cadherin had no significant effect on proliferation of HCA-7 and SK-CO15 cells (Appendix A). In contrast, depletion of this protein significantly upregulated collective IEC migration in the wound healing model (Figure 4C–F). Fixed endpoint wound healing experiments were complimented by live imaging of cell migration in wounded HCA-7 monolayers (Appendix A). Such time lapse imaging revealed significantly higher velocity of cell migration in P-cadherin knockout cell monolayers as compared to control IEC (Appendix A). In addition to examining collective cell migration, we also compared random motility of individual P-cadherin knockout and control HCA-7 cells. Time lapse imaging revealed that loss of P-cadherin increased migrated distance and migration speed of individual IEC (Appendix A). Furthermore, P-cadherin-deficient cells displayed significant decrease in the directionality of cell migration (Appendix A), which may reflect defects in their front-rear polarization. Together these data suggests that P-cadherin regulates wound healing by directly affecting the molecular machinery involved in cell motility rather than influencing it indirectly, via orchestrating junctional remodeling during collective cell migration.

Epithelial cell migration is a multistep process involving coordinated assembly of cell adhesion to extracellular matrix (ECM) and formation of membrane protrusions at the migrating cell edge [66,67]. We next sought to examine whether these molecular events are regulated by P-cadherin in IEC. Loss of P-cadherin expression markedly (more than two-fold) decreased HCA-7 and SK-CO15 cell adhesion to collagen I and fibronectin (Figure 5). Interestingly, the decrease in ECM adhesion was accompanied by accelerated spreading of P-cadherin-depleted cells on collagen matrix (Figure 6). A gain-of-function approach was used to compliment the results obtained with P-cadherin knockout. This approach involved overexpression of P-cadherin in SW620 cells (Appendix A), which have very low endogenous expression of this adhesion protein [68]. In good agreement with the loss-of-function data, overexpression of P-cadherin did not affect cell proliferation (Appendix A), but significantly increased SW620 cell adhesion to the collagen I matrix (Appendix A) and inhibited cell spreading (Appendix A). As neither control, nor P-cadherin-overexpressing SW620 cells, displayed noticeable movement during 48 h of live cell imaging (data not shown), we were unable to examine the effects of P-cadherin overexpression on IEC migration.

### 3.4. Loss of P-Cadherin Promotes IEC Migration via Increase of Src and Rac1 Signaling and NM-II Activation

Next, we sought to elucidate the signaling mechanisms involved in P-cadherin-dependent regulation of IEC migration. Immunoblotting analysis of total cell lysates obtained from wounded control and P-cadherin knockout HCA-7 cell monolayers was carried out to determine the expression and activation status of most essential regulators of cell-ECM adhesion and cell motility/cytoskeletal remodeling. Surprisingly, diminished adhesiveness of P-cadherin-depleted cells was accompanied by neither downregulated expression of major integrins, nor attenuated phosphorylation of integrin-associated scaffolding and signaling molecules (Figure 7A). Oppositely, loss of P-cadherin triggered a profound (~6-fold) activation of cell-ECM adhesion associated Src kinases (Figure 7A). The increase in phosphorylated Src level was also observed in P-cadherin knockout SK-CO15 cells (data not shown), indicating that this is not a cell-specific phenomenon. Consistently, phosphorylation of known molecular targets down-stream of Src, such as focal adhesion kinase (FAK) and paxillin, was increased following P-cadherin knockout (Figure 7A). Furthermore, immunolabeling and confocal microscopy analysis revealed accelerated assembly of focal adhesions containing phosphorylated paxillin in P-cadherin-depleted HCA-7 cells (Appendix A, arrows). In addition to enhancing ECM signaling, P-cadherin depletion affected other signaling mechanisms essential for IEC motility. Specifically, activating phosphorylation of several cytoskeletal proteins such as cofilin and myosin light chains (MLC) was increased, whereas phosphorylation of AKT kinase was decreased in P-cadherin-depleted HCA-7 cells (Figure 7B). In contrast, loss of P-cadherin did not alter expression of different AJ and TJ proteins in HCA-7 cells (Appendix A). These data together with time lapse imaging of individual cell motility (Appendix A) suggest that P-cadherin could regulate IEC migration by mechanisms independent of intercellular junctions.

As the Src family kinases serve as upstream regulators of a number of signaling cascades essential for cell adhesion, migration and cytoskeletal remodeling [69,70], we next investigated whether activation of these kinases play causal roles in the accelerated collective migration of P-cadherin-depleted IEC. Treatment with a selective Src inhibitor, PP2 (20 μM), blocked Src phosphorylation (Figure 8A) and suppressed wound healing in P-cadherin knockout and control HCA-7 cell monolayers (Figure 8B,C), thereby signifying Src activity as a key driver of collective IEC migration. Based on known interactions of Src with cadherin-based adhesion complexes [71,72,73], we rationalized that Src activation could be an upstream signaling event triggered by P-cadherin depletion that influences downstream molecular effectors of cell migration. Interestingly, Src activation in P-cadherin depleted IEC was associated with an increase in cytoskeletal related signaling (Figure 7B) and accelerated cell spreading (Figure 6). It is likely, therefore, that increased Src activity results in remodeling of the cortical actin cytoskeleton, thereby promoting cell spreading and motility.

The Rac1 small GTPase is a key regulator of cell spreading and migration that could be controlled by Src [74,75]. Next, we investigated the roles of Rac1 in accelerated migration of P-cadherin-depleted IEC. The Rac1 activity was significantly (~2.5 fold) upregulated in P-cadherin knockout HCA-7 cells (Figure 9A). Furthermore, treatment with a specific pharmacological Rac inhibitor, EHT 1864 (50 μM) [76], attenuated wound healing in P-cadherin-deficient HCA-7 monolayers (Figure 9B,C).

Our data indicates that in addition to Rac1 activation, loss of P-cadherin also increases activity of RhoA GTPase. This was manifested by the elevated levels of phosphorylated cofilin and di-phosphorylated MLC (Figure 7B), two canonical effector molecules regulated by RhoA signaling [77,78]. As both cofilin and NM-II are known to control cell migration via remodeling and contractility of the actin cytoskeleton [77,79,80], we sought to investigate if activation of these effectors contribute to accelerated motility of P-cadherin-deficient IEC. Pharmacological NM-II inhibitor, blebbistatin [81] (50 μM), attenuated increased wound healing in P-cadherin-depleted IEC monolayers (Appendix A). However, inhibition of cofilin phosphorylation by pharmacological blockage of LIM kinase failed to reverse the accelerated IEC motility caused by P-cadherin knockout (data not shown). Overall, these data indicate that loss of P-cadherin promotes IEC migration by enhancing Src and Rac1 signaling and activating NM-II motors.

### 3.5. Loss of P-Cadherin Induces Transcriptional Reprogramming of IEC

Our findings that the loss of P-cadherin triggers multiple functional and signaling changes in migrating IEC monolayers suggest possible alterations of global molecular networks in P-cadherin knockout cells. To investigate if such global molecular alterations are caused by transcriptional reprogramming, we performed whole genome RNA sequencing analysis of control and P-cadherin-depleted HCA-7 cell lines. This analysis showed profound changes in gene expression of P-cadherin depleted cells with a number of significantly upregulated or downregulated genes (Figure 10A,B; Appendix A). A gene ontology analysis of the RNAseq data revealed that loss of P-cadherin caused upregulation of multiple molecular pathways including those associated with cell migration, membrane protrusion, cell-cell interactions, kinase signaling and responses to inflammatory mediators (Figure 10C and Appendix A). Furthermore several genes upregulated in P-cadherin-deficient IEC lines, including SHROOM2, LARGE1, BMP7, CARD11, LMO7 and AKT3 (Figure 10A,B; Appendix A), encode regulators of cell migration, adhesion and actin cytoskeleton remodeling [82,83,84,85,86]. Overall, our data suggests that P-cadherin modulates collective cell migration by controlling multiple molecular and signaling pathways in IEC.

## 4. Discussion

P-cadherin is a member of the classical cadherin family with limited tissue-specific expression under homeostatic condition and poorly understood functions in development and diseases. Our study shows marked induction of P-cadherin expression in the colonic epithelium of UC and CAC patients (Figure 1). We demonstrate for the first time that P-cadherin expression attenuates post-colitis recovery but is not essential for colitis-induced colon cancer growth in vivo (Figure 2 and Figure 3). Furthermore, our study revealed potent anti-migratory activity of P-cadherin in cultured IEC that is underlined by regulation of cell-ECM adhesion and control of multiple molecular and signaling pathways essential for cell motility.

The existing literature presents a conflicting view for the roles of P-cadherin in regulating motility of different epithelial and cancer cells, by reporting either promigratory [34,36,43,87,88], or antimigratory [89,90,91,92,93] activities of this adhesion protein. Two major reasons for such a diverse and context dependent regulation of cell motility by P-cadherin could be envisioned. One is a known functional interplay between P-cadherin and other classical cadherins expressed in epithelial and cancer cells. For example, the effects of P-cadherin on cancer cell motility are modulated by the expression of E-cadherin [36,37,42]. Depending on expression levels of other homologous cadherins, P-cadherin can either promote or destabilize cell-cell adhesions thereby altering collective cell responses in migrating epithelial monolayers. Additionally, P-cadherin and E-cadherin compete for key AJ scaffolding proteins, such as p120 catenin [94], which is an essential regulator of cell migration [88,95]. In inflamed intestinal mucosa and well-differentiated colon cancer, P-cadherin is co-expressed with E-cadherin (Figure 1) and could either antagonize or synergize with E-cadherin function. In more advanced metastatic colon cancer undergoing EMT, P-cadherin likely cross-talks with mesenchymal type N-cadherin and cadherin-11. Another possible reason for the diverse roles of P-cadherin in regulating cell motility may be attributed to differences in the cytoskeletal organization in different epithelial and cancer cells. P-cadherin dependent transmission of contractile/traction cytoskeletal forces between neighboring cells acts as essential regulator of collective cell movement [89,96,97]. As the levels of cytoskeleton-generated mechanical stresses vary significantly between different epithelial and cancer cells, this could result in different functional outcomes for P-cadherin-dependent mechanotransduction.

The results of the present study (Figure 4 and Appendix A), together with published data [44], suggest that in transformed differentiated IEC lines (HCA-7, SK-CO15, HT-29), P-cadherin acts as a negative regulator of cell migration. This antimigratory effect is likely driven by the interplay between two mechanisms: a P-cadherin-induced increase in cell-ECM adhesion (Figure 5 and Appendix A) and inhibition of membrane protrusions. There is a complex relationship between the velocity of cell migration and strength of cell-ECM adhesion, with both weak and strong ECM adhesion inhibiting cell motility [98,99,100]. Weaker ECM adhesion of P-cadherin depleted IEC could reflect the accelerated remodeling of matrix adhesion complexes contributing to the enhanced collective cell migration. Our finding of inhibited ECM adhesion in P-cadherin deficient IEC is consistent with previously published evidence obtained in breast, gastric and ovarian cancer cells [101,102,103], although the mechanisms of such reduced adhesion remain to be explored. Lower ECM adhesion of P-cadherin-depleted breast and ovarian cancer cells is accompanied by the decreased expression of α6, β1 and β4 integrin subunits [102,103]. Furthermore P-cadherin in myoblast cells was shown to affect ECM composition by regulating expression of an important ECM protein, decorin [104]. However, we did not find evidence for similar mechanisms in IEC, as P-cadherin knockout did not inhibit expression of key colonic integrins (Figure 7), or decorin (data not shown). Given a well-known functional crosstalk between cadherin-based AJs and integrin-based ECM adhesion [105,106], it is possible that P-cadherin regulates IEC attachment to the matrix by modulating localization and activation status of different integrin complexes at the plasma membrane. 

Our study revealed several signaling events regulated by P-cadherin in migrating IEC. One important event involves activity of Src kinases. Indeed, loss of P-cadherin resulted in Src activation in HCA-7 (Figure 7) and SK-CO15 (data not shown) cells. Furthermore, pharmacological inhibition of Src activity reversed the accelerated collective migration of P-cadherin deficient HCA-7 cells (Figure 8). The described inhibition of Src activity by P-cadherin is consistent with published studies that demonstrate Src localization at epithelial junctions and describe either negative or positive regulation of Src activity by classical and desmosomal cadherins [72,73,107,108,109]. Furthermore, Src is a well-recognized regulator of cell migration that affects multiple molecular events in motile cells [70,110,111]. Surprisingly, Src activation in P-cadherin-depleted IEC did not result in stimulation of ECM adhesion, despite activation of downstream effector molecules such as FAK and paxillin (Figure 7) and acceleration of focal adhesion assembly (Appendix A). Such paradoxical outcomes of Src activation suggest that high Src activity could trigger dynamic instability of ECM adhesions resulting in accelerated migration of P-cadherin-deficient IEC. Mechanisms underlying such Src-dependent destabilization of ECM adhesions likely involve regulation of the actomyosin cytoskeleton [70,110,111].

An important signaling event controlled by P-cadherin in migrating IEC cells involves regulation of Rho family of small GTPases, such as Rac1 and RhoA. Our data demonstrates that P-cadherin inhibits cell spreading (Figure 6 and Appendix A), which is indicative of antagonizing Rac1 function. Furthermore, the loss of P-cadherin increased the level of active Rac1 and pharmacological inactivation of this small GTPase reversed accelerated collective migration of P-cadherin deficient cells (Figure 9). Likewise, P-cadherin knockout increased phosphorylation of cofilin and MLC (Figure 7), which could be consequential of enhanced RhoA signaling. By regulating different members of the Rho GTPase family, P-cadherin can control different stages of IEC migration. Specifically, P-cadherin likely regulates Rac-dependent membrane protrusions at the migrating cell edge and controls Rho-NM II-dependent mechanical forces at epithelial junctions and cell-ECM adhesions [33]. The mechanisms underlying P-cadherin-dependent modulation of Rho GTPase activity in IEC remain to be investigated. There is a possibility that P-cadherin regulates Rac1 and RhoA activity indirectly, via Src signaling. Indeed, P-cadherin depleted HCA-7 cells are characterized by co-activation of Src and Rac1 (Figure 7 and Figure 9). Furthermore, activated Src was shown to modulate Rac1 and RhoA signaling in migrating cancer cells [74,75,112,113].

The observed activation of Src and Rho GTPases appears to be a part of the global rearrangement of molecular/signaling networks caused by P-cadherin depletion in IEC. Indeed, our RNA sequencing analysis revealed an extensive transcriptional reprogramming of P-cadherin-deficient HCA-7 cells (Figure 10 and Appendix A; Appendix A). This transcriptional reprograming correlates well with the described functional effects of P-cadherin knockout and could play a causal role in these responses. For example, upregulation of molecular pathways linked to cell motility is consistent with the increased migration of P-cadherin-deficient IEC (Figure 4 and Appendix A). Likewise, the observed modulation of protein kinase signaling, and protein phosphorylation pathways correspond well to the altered phosphorylation of Src and some cytoskeletal proteins (Figure 7). Of note, our RNA sequencing data is in agreement with two previous microarray analyses of P-cadherin deficient and overexpressing breast cancer cells that demonstrated altered mRNA levels of proteins regulators to cell motility, matrix adhesion and the actin cytoskeleton [36,92]. The described P-cadherin-dependent modulation of gene expression is not surprising, given well-known association of cadherin-based AJs with several key regulators of gene expressions. The best studied examples of such AJ-associated transcriptional regulators include β-catenin, p120 catenin and Yes-associated protein [19,114,115]. 

In contrast to the strong evidence implicating P-cadherin in the regulation of IEC matrix adhesion, migration, and mucosal repair, we found little evidence that this cadherin is essential for colon cancer development. Indeed, depletion of P-cadherin affected neither colonic epithelial cell proliferation in vitro (Appendix A) nor colonic tumor growth under inflammatory conditions in vivo (Figure 3). Interestingly, loss of P-cadherin resulted in development of higher-grade colonic tumors in mice (Figure 3E), which suggests that P-cadherin plays a role in controlling the phenotypic plasticity of CAC. Lack of effect of P-cadherin knockout on CAC burden in mice is consistent with previous studies showing that overexpression of P-cadherin did not affect epithelial cell proliferation in mice. Thus, transgenic mice with IEC-specific overexpressing murine P-cadherin driven by the L-FABP promoter did not show altered cell proliferation in the small intestinal or colonic epithelium under normal conditions [30]. Furthermore, indomethacin induced mucosal injury did not result in the hyperproliferative response in P-cadherin overexpressing IEC [30]. Mammary tumorigenesis was studied in another transgenic mouse model with overexpression of human P-cadherin in mammary epithelial cells. In this study, breast cancer was induced by the *neu* oncogene [116]. Interestingly, no effects of P-cadherin overexpression on the average number and the size of developed tumors were observed. Therefore, our data concurs with existing literature and argues against the role of P-cadherin in regulating colon tumor growth in vivo.

The results obtained in our study set the stage for future works aimed at understanding the physiological roles of P-cadherin upregulation in injured, inflamed, or neoplastic colonic mucosa. Such P-cadherin upregulation has pro-adhesive and anti-migratory effects and therefore could impede the excessive and uncoordinated IEC movement. The pathophysiological consequences for induction of such an anti-reparative molecule as P-cadherin remain puzzling. It appears to be detrimental for restoration of acutely injured intestinal mucosa but could be beneficial during multiple rounds of repair in chronically inflamed intestine by preventing excessive IEC loss and orchestrating cell behavior at topographically different regions of the remodeling epithelial layer. Furthermore, attenuated migration of P-cadherin expressing IEC cells is expected to decrease the crypt to villous epithelial cell transit in the inflamed colonic mucosa, which may affect stem cell behavior and differentiation. Finally, induction of P-cadherin expression during colon cancer development could serve as an adaptive antimetastatic response antagonizing either classical EMT or tumor cell invasion/dissemination. Another possible role of P-cadherin upregulation in IEC is alluded to by our RNA sequencing analysis that shows alterations in pathways related to chemokine and cytokine receptor signaling and responses in P-cadherin-depleted IEC cells (Appendix A). This indicates that P-cadherin could modulate cross-talk between IEC and immune cells in either inflamed mucosa, or in the tumor microenvironment. While recent reports highlight P-cadherin as a possible biomarker for local immune surveillance in certain cancers [117,118], nothing is known about the P-cadherin function in regulating the immune environment, which could be an important focus for future investigations. 

In conclusion, our study unravels novel roles for P-cadherin, an important junctional protein upregulated in the colonic mucosa of UC and CAC patients, in regulating migration and signaling in IEC. We discovered that P-cadherin acts as a negative regulator of collective cell migration in vitro and mucosal restitution in vivo. By contrast, we did not find evidence that P-cadherin controls proliferation of human IEC or tumor growth in the mouse model of CAC. The observed anti-migratory effects of P-cadherin involve regulation of cell-ECM adhesion and cell spreading by altering signaling of the Src family kinases, and the Rac1 GTPase and NM-II motors. Additionally, P-cadherin was found to be a modulator of the global transcriptional program in IEC and specifically, molecular pathways related to cell migration, regulation of kinase activity and cytokine/chemokine responses. Therefore, P-cadherin could serve as a potential therapeutic target to accelerate intestinal restitution and increase efficiency of immunotherapeutic interventions in colorectal cancer.

## Figures and Tables

**Figure 1 cells-11-01467-f001:**
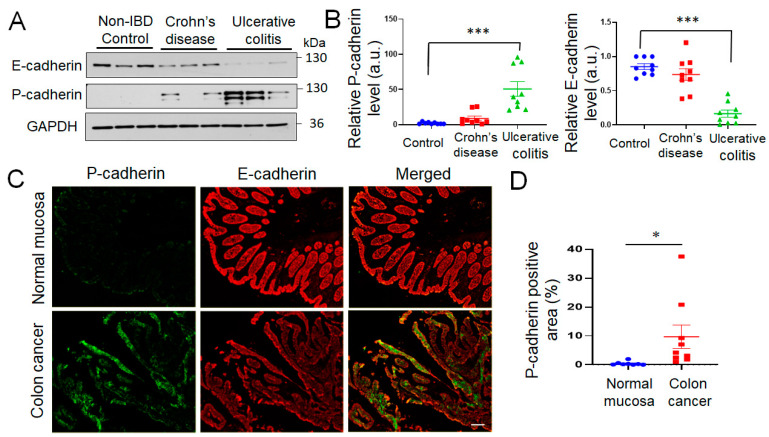
Increased expression of P-cadherin protein in colonic mucosa of ulcerative colitis and colitis-associated colon cancer patients. (**A**,**B**) Immunoblotting analysis of P-cadherin and E-cadherin expression in isolated colonic epithelial cells of Crohn’s disease, ulcerative colitis patients and non-IBD controls. Representative immunoblots (**A**) and densitometric quantification of P-cadherin and E-cadherin protein levels (**B**) are shown. Means ± SE (*n* = 9 in each experimental group); *** *p* < 0.005. (**C**,**D**) Dual immunofluorescence labeling of P-cadherin (green) and E-cadherin (red) in whole thickness tissue sections of colitis-associated colon cancer patients and normal controls. Representative fluorescence microscopy images (**C**) and quantification of P-cadherin-positive area of the colonic mucosa (**D**) are shown. Means ± SE (*n* = 8 and 9 for the control the CAC groups, respectively); * *p* < 0.05. Scale bar, 100 μm.

**Figure 2 cells-11-01467-f002:**
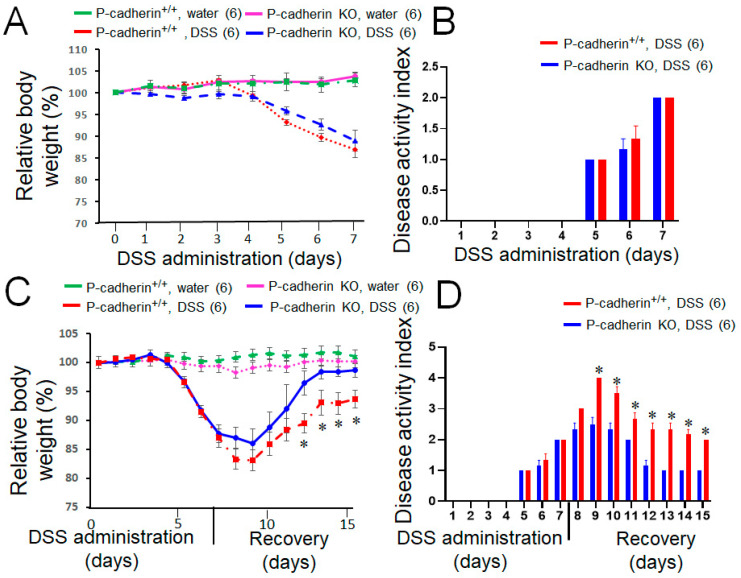
Loss of P-cadherin does not affect development of acute DSS colitis but promotes post-colitis recovery. P-cadherin knockout (KO) mice and their control littermates (+/+) were subjected to acute DSS colitis (3% DSS in drinking water) for 7 days (**A**,**B**) or exposed to 3% DSS for 7 days and allowed to recover after DSS withdrawal (**C**,**D**). Body weight (**A**,**C**), and disease activity index (**B**,**D**) were recorded daily. Mean ± SE. Number of animals in each group is shown in parentheses. * *p* < 0.05. Data are representative of two independent experiments.

**Figure 3 cells-11-01467-f003:**
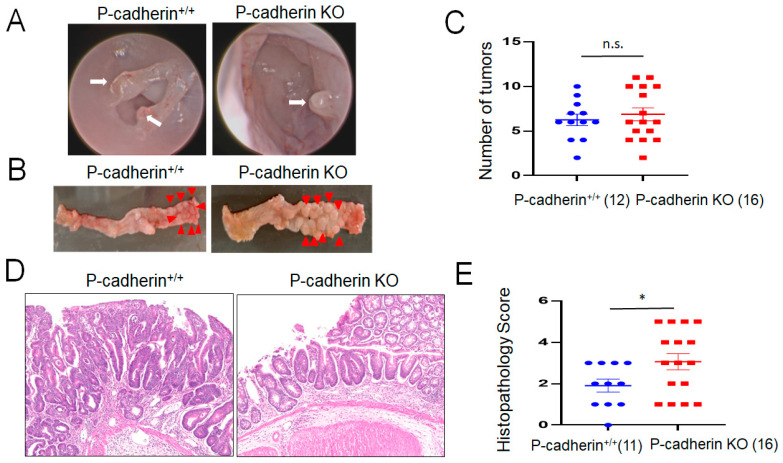
Development of colitis-associated colon cancer is not altered in P-cadherin knockout mice. Control and P-cadherin KO mice were injected intraperitoneally with AOM and subjected to three cycles of DSS exposure and recovery. Mice were euthanized and their colonic polyps were counted and examined. (**A**) An endoscopic view of the colonic polyps (arrows). (**B**) Overall view of the dissected colonic segments with tumors (arrowheads). (**C**) Number of developed colonic tumors. (**D**) Representative hematoxylin and eosin staining of the colonic sections showing high grade neoplasia and low grade neoplasia in control and P-cadherin knockout mice, respectively. (**E**) The histopathology score. Number of animals of each group is shown in parentheses. * *p* < 0.05; n.s., not significant, *p* = 0.594. Data were combined from two independent experiments.

**Figure 4 cells-11-01467-f004:**
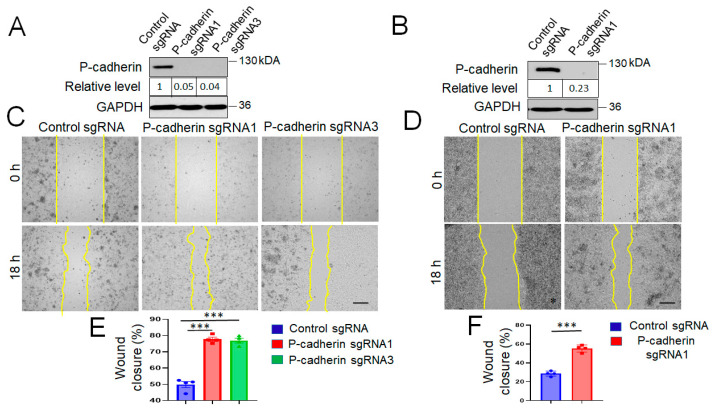
Knockout P-cadherin promotes intestinal epithelial cell migration: (**A**,**B**) Immunoblotting analysis shows the efficiency of CRISPR/Cas9-mediated P-cadherin knockout in HCA-7 and SK-CO15 cells. (**C**,**D**) Representative wound images and (**E**,**F**) quantification of cell migration at 18 h post-wounding in control and P-cadherin-depleted HCA-7 and SK-CO15 cell monolayers. Means ± SE (*n* = 4); *** *p* < 0.005. Scale bars, 100 μm. The results shown are representative of three independent experiments.

**Figure 5 cells-11-01467-f005:**
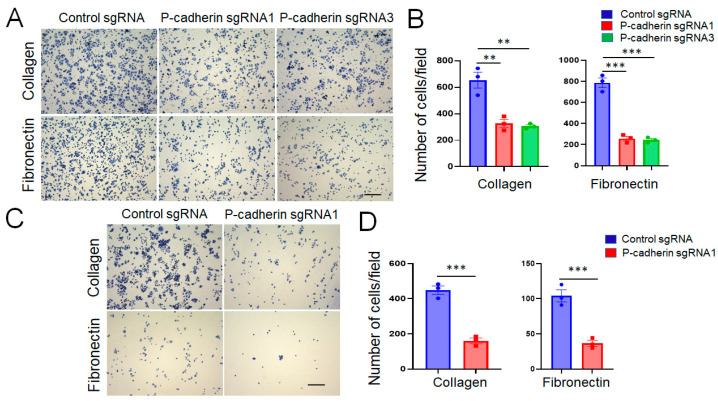
Loss of P-cadherin expression decreases intestinal epithelial cell adhesion to extracellular matrix. Control and P-cadherin-depleted HCA-7 (**A**,**B**) and SK-CO15 (**C**,**D**) cells, were plated on either collagen I, or fibronectin-coated plates and adherent cells were counted 1 h after plating. Representative images (**A**,**C**) and quantification of adhered cells (**B**,**D**) are shown. Means ± SE (*n* = 4); ** *p* < 0.01, *** *p* < 0.005. Scale bars, 100 μm. The results shown are representative of three independent experiments.

**Figure 6 cells-11-01467-f006:**
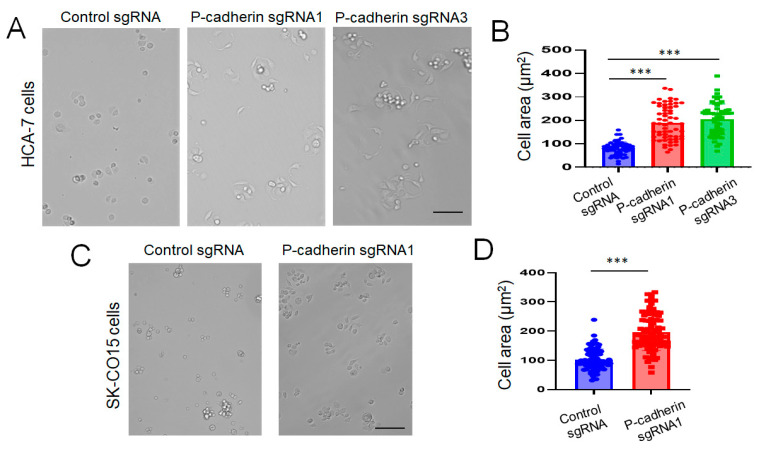
Loss of P-cadherin expression promotes intestinal epithelial cell spreading. Control and P-cadherin-depleted HCA-7 (**A**,**B**) and SK-CO15 (**C**,**D**) cells were plated on collagen I coated plates, allowed to spread for 1 h and imaged using bright field microscopy. Representative images of spread cells (**A**,**C**) and quantification of the cell area (**B**,**D**) are shown. Means ± SE (*n* = 60 and 90 for HCA-7 and SK-CO15 cells, respectively); *** *p* < 0.005. Scale bars, 100 μm. The results shown are representative of two independent experiments.

**Figure 7 cells-11-01467-f007:**
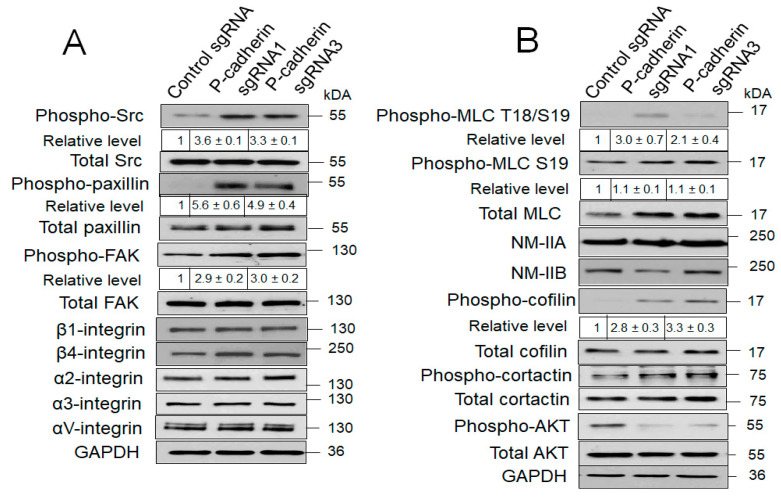
P-cadherin knockout affects matrix adhesion and actin cytoskeleton-related signaling. Immunoblotting analysis of expression and phosphorylation of different proteins involved in cell-matrix adhesion (**A**) and actin cytoskeleton remodeling (**B**) in control and P-cadherin knockout HCA-7 cells.

**Figure 8 cells-11-01467-f008:**
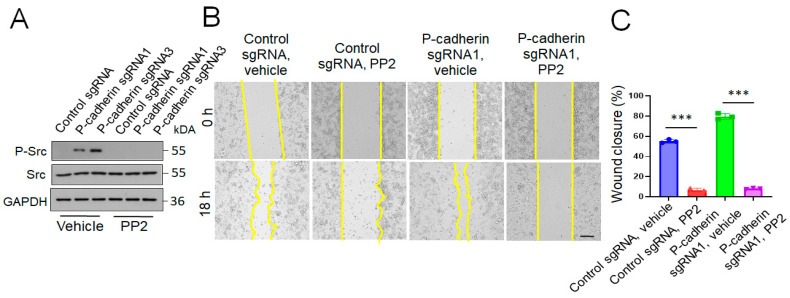
Src inhibition reverses the accelerated wound healing in P-cadherin deficient IEC monolayers. Control and P-cadherin knockout HCA-7 cell monolayers were wounded and allowed to migrate in the presence of either vehicle, or Src inhibitor, PP2 (20 μM). Immunoblotting analysis demonstrates inhibition of Src phosphorylation by PP2 (**A**). Representative pictures of wounded monolayers (**B**) and quantification of wound closure (**C**) are shown. Means ± SE (*n* = 4); *** *p* < 0.005. Scale bar, 100 μm. The results shown are representative of three independent experiments.

**Figure 9 cells-11-01467-f009:**
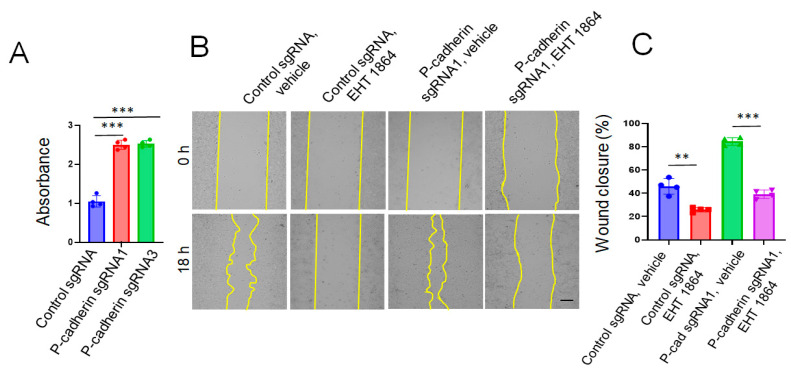
Rac1 activation mediates the increased wound healing in P-cadherin deficient IEC monolayers. (**A**) G-LISA assay shows Rac1 activation in P-cadherin knockout HCA-7 cells. Control and P-cadherin knockout HCA-7 cell monolayers were wounded and allowed to migrate in the presence of either vehicle, or Rac1 inhibitor, EHT 1864 (50 μM). Representative pictures of wounded monolayers (**B**) and quantification of wound closure (**C**) are shown. Means ± SE (*n* = 4); ** *p* < 0.01, *** *p* < 0.005. Scale bar, 100 μm. The results shown are representative of two independent experiments.

**Figure 10 cells-11-01467-f010:**
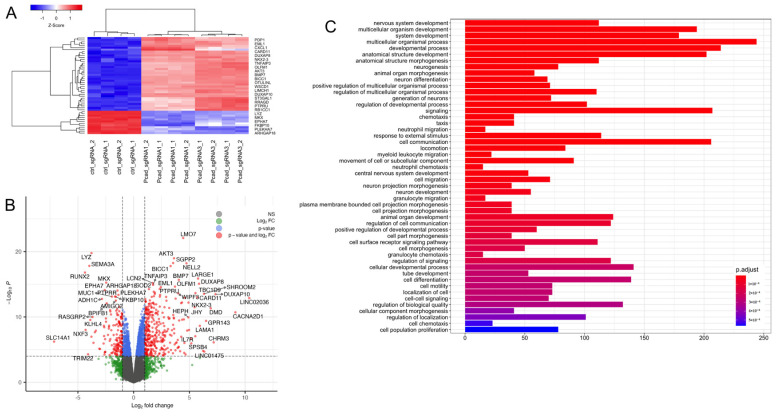
Effect of P-cadherin knockdown on the transcriptional network in intestinal epithelial cells. (**A**) Heatmap showing top 50 differentially expressed genes (fold change > 2) in two P-cadherin knockout HCA7 cell lines. (**B**) Volcano plot showing differentially expressed genes in P-cadherin knockout cells compared to control. (**C**) Pathways mapped to differentially expressed genes in P-cadherin knockout cells.

## Data Availability

The RNA sequencing data have been deposited in Gene Expression Omnibus. The GEO accession number is GSE199859.

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
