# Peer review of "P-Cadherin Regulates Intestinal Epithelial Cell Migration and Mucosal Repair, but Is Dispensable for Colitis Associated Colon Cancer"

_cells, 2022, doi:10.3390/cells11091467_

Round 1

Reviewer 1 Report

The current study reported that P-cadherin as a negative regulator of IEC motility in vitro and mucosal repair in vivo.  P-cadherin is dispensable for IEC proliferation and CAC development. Previous studies addressing the involvement of P-cadherin in colon cancer cell adhesion and migration reported conflicting cell-specific phenotypes and did not provide mechanistic insights  into P-cadherin modulation of IEC motility. Functional roles of P-cadherin in the regulation of intestinal mucosal restitution and colonic tumorigenesis in vivo remain poorly understood. The current study provided new insights into the field using bot in vitro and in vivo models. The overall experimental design was thoughtful. The general conclusion is supported by the data. The reviewer only have minor comments for the authors to improve the manuscript:

  1. Fig. 3 needs to show n of each group, mice age and gender, statistic methods, ad p values.
  2. Did the authors test the changes of beta-catenin in the colon cancer model?
  3. Please consider further discussion on the potential synergetic role of p-cad with other cadherins.
  4. A working model of p-cad in IBD and colon cancer will help readers better understand the current findings.

Author Response

Reviewer 1.

We thank the reviewer for stating that “The current study provides new insights into the field using bot in vitro and in vivo models. The overall experimental design was thoughtful and the general conclusion is supported by the data.” and for providing the valuable comments, which are addressed below:

Minor Concerns:

Comment 1: Fig. 3 needs to show n for each group, mice age and gender, statistic methods and p value.
Response: As suggested by the reviewer we indicated the n for each animal group in Figure 3 (number presented in parenthesis) and included the p values in the Figure legend. Animal age was 8-10 weeks at the beginning of the study and we used approximately equal numbers of male and female mice in each experimental group. We indicate this in the Methods section. Statistical analysis of the animal data was performed using two-tailed unpaired Student t-test, which showed statistically significant difference in tumor histology, but not tumor number between P-cadherin knockout mice and their wild type controls. We now better describe the statistical analysis of all in vivo and in vitro experiments in the appropriate Methods section.

Comment 2: Did the authors test the changes of beta catenin in the colon cancer model?

Response: Since we did not find significant differences in the tumor burden between P-cadherin knockout mice and their wild type controls, we did not examine beta-catenin or other molecular pathways related to tumor cell proliferation in these animals. However, we analyzed expression of beta-catenin and other protein components of the cadherin-catenin complexes in control and P-cadherin knockout human colon cancer cells, but did not find the effects of loss of P-cadherin expression on the levels of either E-cadherin, or associated alpha- beta- and p120-catenins. We now present these data in a new Figure S7 in the revised manuscript.

Comment 3: Please consider further discussion on the potential synergistic role of P-cadherin and other cadherins.
Response: As suggested by the reviewer we discussed potential interplay between P-cadherin and other cadherins in intestinal epithelial and colon cancer cells in the Discussion section of the revised manuscript.

Comment 4: A working model of P-cad in IBD and colon cancer will help readers better understand the current findings.

Response: We appreciate the reviewer’s comment, however, we do not have capacity of drawing meaningful research diagrams in the lab and due to the short time allowed for the manuscript revision we were unable to get help from the Cleveland Clinic Art and Illustration Department.

Reviewer 2 Report

The manuscript authored by Nayden G, Naydenov and colleagues is an elegant study which demonstrates a role for P-cadherin as a negative regulator of intestinal cell migration and mucosal repair. First, the authors determined that P-cadherin expression was upregulated in the colonic epithelium of human IBD patients and colitis-associated colon cancer (CAC) tissue. Then, P-cadherin knockout mice and wild type littermates were subjected to a mouse model of acute DSS colitis, recovery after DSS-induced colitis or AOM/DSS-induced CAC. Authors found that loss of P-cadherin resulted in faster recovery after colitis but did not affect the severity of acute DSS colitis or tumorigenesis. By employing human intestinal cell lines, authors showed that loss of P-cadherin led to increased epithelial cell migration post-wounding and cell spreading but did not affect cell proliferation. Pharmacological inhibitors revealed that loss of P-cadherin resulted in the activation of Src kinases, Rac1 and myosin II. Authors also conducted a whole genome RNA sequencing analysis in P-cadherin knockout cells compared to control cells which corroborated an alteration of pathways regulating cell migration, adhesion and actin cytoskeleton remodeling. Overall, the experiments are well-described, and the manuscript is well written. The conclusions are based on the results presented. However, I have specific concerns that I feel should be addressed to strengthen the manuscript.

1- In Figure 1C, co-staining with markers of human colon tumor/ cell proliferation should be added (such as b-catenin, CD44 or Ki67). Also, it would be helpful if the upregulation of P-cadherin expression was also shown in inflamed mouse colon in the animal model of colitis and CAC.

2- In Figure 2, authors should add representative H&E images of colon Swiss rolls to help appreciate the extend of injury and epithelial restitution. Staining of the tissue with an intestinal epithelial specific marker (such as E-cadherin or Villin) should be considered.

3- Authors mentioned “a cadherin switch” which is characterized by downregulation of E-Cadherin and upregulation of P-cadherin. It is likely that a cadherin switch might destabilize cell-cell cohesion and collective migration as a sheet of cells. In the movies provided, it seems that cells at the leading-edge of the monolayer lacking P-cadherin show weakened cell-cell cohesion and impaired collective migration compared to control. There are no experiments that examined whether the levels of P-cadherin in the cell lines resulted in change in the expression of E-cadherin or other intercellular junctional proteins which play a key role in regulating cell-cell cohesion. These experiments should be included.

4- It would be helpful if immunofluorescence staining of focal adhesion complexes (including p-FAK and p-paxillin) was included to complement the cell spreading and migration assays.

5-Authors should consider including key signaling studies with an additional cell line such as SKCO15 cells. A non-cancerous state could be also considered by using primary culture of mouse intestinal epithelial cells derived from the P-cadherin KO versus control mice.

Author Response

Reviewer 2.

We thank the reviewer for stating that “the experiments are well-described and the manuscript is well written” and providing insightful comments that we address below:

Comment 1: In Figure 1C co-staining with markers of tumor cell proliferation should be added. Also in would be helpful in upregulated P-cadherin expression is shown in in flamed mouse colon in the animal model of colitis and CAC.

Response: We appreciate this comment, however, given our data that P-cadherin does not regulate colon cancer growth in vivo and colon cancer cell proliferation in vitro, we believe that the suggested co-immunolabeling of P-cadherin with proliferation markers in human CAC would not provide any additional mechanistic insights and could be distractive from P-cadherin-dependent regulation of cell migration.

A methodological problem precluded us from examining P-cadherin expression in inflamed or neoplastic murine intestinal mucosa. While many commercially available antibodies detect P-cadherin expression and localization in human tissues, we observed that they do not work in mouse tissues. After prolonged trials and failures we found only one goat polyclonal antibody that recognizes P-cadherin in mouse tissues by immunoblotting (see Figure S1). Unfortunately, we cannot get good-P-cadherin immunofluorescence labeling with this antibody in mouse intestinal samples. 

Comment 2: In Figure 2, authors should add representative H&E images of colon Swiss rolls to appreciate the extent of injury and restitution. Staining with intestinal epithelial specific markers should be considered.

Response: We do not have Swiss rolled colons from these experiments since this technique consumes too much tissue that is needed for other analysis. Instead we use serial sections of the distal colon. As suggested by the reviewer we now show representative colonic sections of the animals after DSS recovery in a new Figure S2. Since these H&E sections clearly show the extent of colonic epithelial restoration in wild type and P-cadherin null mice, we do not believe that additional immunolabeling for epithelial markers is necessary. We hope the reviewer agrees with such assessment.

Comment 3: There are no experiments that examined whether levels of P-cadherin in the cell lines results in change in the expression of E-cadherin or other intercellular junctional proteins which play a key role in regulating cell-cell cohesion. These experiments should be included.

Response: As suggested by the reviewer we now included immunoblotting analysis of control and P-cadherin knockout HCA-7 cells that shows no effect of P-cadherin knockout on expression of different AJ and TJ proteins. This data is presented in a new Figure S7.

Comment 4: It would be helpful if immunofluorescence labeling of focal adhesion complexes (P-FAK and p-PAX) is included to complement the adhesion and spreading assay.

Response: As suggested by the reviewer we now present immunolabeling of focal adhesions in spreading control and P-cadherin knockout HCA-7 cells in a new Figure S6 of the revised manuscript.  

Comment 5. Authors should include key signaling studies with an additional cell line such as SKCO15. A non-cancerous state should also be considered using primary culture of mouse intestinal epithelial cells.

Response: We have data demonstrating significant increase in Src phosphorylation, which a key signaling event triggered by P-cadherin knockout in SK-CO15 cells. This is consistent with our results obtained in HCA-7 cells (Figure 7). We now mention such Src activation in the SK-CO15 cells as ‘data not shown’ in the revised manuscript. We agree that it is important to investigate P-cadherin related signaling and functional events in non-tumorigenic primary epithelial cells, but this is not trivial and time consuming task that will be pursued in a follow up study.